# QUERY OPTIMIZATION DETECTION TRANSFORMER FOR SMALL OBJECTS IN REMOTE SENSING IMAGES

## ABSTRACT

Object detection in remote sensing images is a challenging task. Remote sensing images contain substantial background noise and complex contextual information, which weakens the feature representation of small objects, making detection difficult. To solve these problems, a detection Transformer for small objects in remote sensing images is proposed, called QO-DETR. Specifically, to enhance the feature representation of small objects, a query proposal generation module is designed to select queries based on multi-class classification scores. These queries provide the initial position embeddings for object queries in the decoder, enabling the decoder's attention mechanism to focus on object regions. To improve the model's robustness to noise, a group denoising module is designed to add noise into decoder queries during training, enhancing the network's ability to reconstruct object features from noise. To accurately locate small objects, a query cascade refinement strategy is designed, and each decoder layer refines anchor parameters under the guidance of preceding layers to achieve spatial alignment between the anchor and the object. Experiments have been carried out on DIOR and AI-TOD. The AP and $AP_S$ on DIOR reach 51.3% and 13.4%, respectively, while on AI-TOD, they reach 23.6% and 30.1%. QO-DETR shows superior performance in detecting small objects.

## 1 INTRODUCTION

With the rapid development of Earth observation technology, the spatial, temporal and spectral resolutions of remote sensing satellites have been significantly improved, and a large number of remote sensing images are now publicly accessible. Object detection in remote sensing images is an important task in the field of remote sensing and has received extensive research attention in civilian and military fields. In remote sensing images with large sizes and fields of view, objects often only occupy a small part of the image, and even the most advanced general detection algorithms have difficulty in accurately locating these objects in complex backgrounds. Therefore, achieving accurate and efficient detection of small objects in remote sensing images, such as vehicles and pedestrians, has become a key step in improving the level of remote sensing applications (Hua & Chen, 2023).

Remote sensing images are taken from high-altitude viewpoints at different ground sampling distances, covering a wide area of the earth's surface. Therefore, objects in the image space are often distributed in arbitrary directions, and there are huge scale differences between objects. Many methods of remote sensing object detection focus on the arbitrary orientation properties of objects. Some learn rotated object representations (Dai et al., 2022)(Yang & Yan, 2020), and some learn rotation-invariant features of objects. In order to solve the problem of large scale variations, some methods construct size-specific detectors to allow features at different levels to independently detect objects of corresponding scales; some methods (Lv et al., 2022)(Xiao et al., 2023a) fuse hierarchical features to capture detailed information and rich semantic information to obtain multi-scale feature representations. In addition, there is a multi-scale anchor box generation method that uses fixed anchor boxes (Yang & Yan, 2020) as priors or uses dynamic anchor boxes (Xiao et al., 2023b) to adaptively learn anchor box coordinates. Remote sensing images also contain complex contexts and a large amount of background noise. Some methods (Yang et al., 2019)(Liu et al., 2022) use attention mechanism to enhance the feature response of the object while suppressing background noise. Some methods (Gong et al., 2020)(Li et al., 2022) assist detection by mining local context information or global context information. Although deep learning methods have made significant progress in remote

sensing object detection, they perform poorly in small object detection. Weak feature representation is the main reason for the poor performance of small object detection and it comes from many aspects (Cheng et al., 2023). First, small objects themselves cover fewer image pixels, resulting in less feature information. Second, the downsampling operation of deep neural networks will further lead to the loss of feature information of small objects. In addition, the regional features of small objects can be easily submerged by the background and other instances. Constructing multi-scale representations (Liu et al., 2016) is the most common method to enhance the feature representation of small objects. Among them, Feature Pyramid Networks (FPN) (Lin et al., 2017a) and its extensions (Yang et al., 2022) are widely used in small object detection algorithms. Attention-based methods [8] highlight small object regions and suppress background regions by assigning different weights to different regions of the feature map. Context modeling methods (Rekavandi et al., 2023) mine the spatial relationship between small objects and the environment or other objects to provide auxiliary feature representations for small objects. Region proposal methods (Ren et al., 2017) first extract the regions containing the objects and then perform detection only on these regions, thereby filtering out additional noise information to improve detection. Super-resolution methods (Bashir & Wang, 2021) restore or reconstruct the original low-resolution features to a higher resolution to obtain more details about the small objects. In addition, the performance of detectors based on deep learning is closely related to the image data itself. For small object detection tasks, general sampling strategies usually fail to provide enough positive samples, which impairs the final performance. Sample-oriented methods (Lin et al., 2017b)(Zhang et al., 2020a)(Wang et al., 2021a) increase the number of small objects by data augmentation or designing optimal allocation strategies to achieve sufficient network learning samples.

Currently, there is little research on small object detection in remote sensing images. In addition to the large scale variation of objects, complex context information and high background noise, small objects are usually sparsely distributed in large-scale and large-field-of-view remote sensing images, which further increases the difficulty of detection. The Transformer-based detector uses attention mechanism to model the interactions between pairs of positions in the input image, which has strong capabilities of context capture, and outperforms CNNs-based detectors in small object detection (Rekavandi et al., 2023). DETR (Carion et al., 2020) uses learnable queries to explore the image features output by the Transformer encoder and performs set-based bounding box prediction, discarding the hand-designed components that encode prior knowledge. However, DETR's training convergence is slow and its performance is relatively poor when detecting small objects. Two-stage Deformable DETR (Zhang et al., 2020b) introduces deformable attention and region proposals to optimize the above problems.

Inspired by the two-stage Deformable DETR, an end-to-end Transformer for small object detection in remote sensing images is designed, called QO-DETR. By optimizing the object query in the Transformer decoder, it alleviates the impact of weak small object feature representation and high background noise in remote sensing images on small object detection performance. QO-DETR represents the Transformer query as a combination of 4-D dynamic anchor boxes and content features. In order to enhance the feature representation of the object, a Query Proposal Generation (QPG) module is designed to select queries based on scores of multi-class classification and provide initial position embedding for the object query of the decoder, thereby providing position constraints for feature aggregation in the attention module of the decoder, so that the cross-attention module focuses on the object region, which not only obtains features with stronger object representation capabilities, but also helps the network locate the object region. To accurately locate small objects, a Query Cascade Refinement (QCR) strategy is designed for the decoder. Each decoder layer refines the query anchor box parameters layer by layer under the guidance of the adjacent layer to achieve spatial alignment between the anchor box and the object. In order to improve the model's robustness to noise, a Group Denoising (GD) module is designed to introduce noise to candidate queries during training, enhance the network's ability to reconstruct object features from noise, and reduce the interference of noise on bipartite graph matching.

In summary, this paper has the following contributions:

(1) We design an end-to-end Transformer detection network for small objects in remote sensing images. We demonstrate that QO-DETR has excellent performance in the small object detection task in remote sensing images on the DIOR dataset and AI-TOD dataset, which promotes the application of Transformer in the field of small object detection in remote sensing images.

(2) QPG is designed to provide initial priors for the decoder's object query. Together with QCR, it guides the decoder's attention module to focus on object region features more quickly and accurately and gradually align the query anchor box with the object.

(3) GD is designed to introduce noise into the decoder query during training, which enhances the model's ability to reconstruct object features from noise and improves the model's robustness to noise.

## 2 RELATED WORK

### 2.1 OBJECT DETECTION IN REMOTE SENSING IMAGES

Object detection in remote sensing images is usually an oriented object detection task, which uses a rotated bounding box to represent the object position. AO2-DETR(Dai et al., 2022) is an end-to-end Transformer for arbitrary-direction object detection. It proposes a rotation proposal generation mechanism to provide better position priors for aggregating features to regulate cross-attention in the transformer decoder. It also designs an adaptive rotation region proposal refinement module to extract rotation-invariant region features and eliminate the misalignment between region proposals and objects. To address the issue of large variations in object size, MFAF(Lv et al., 2022) uses adaptive fusion in which a multi-scale feature integration module and a spatial attention weight module achieve multi-scale features. It then uses a detail enhancement module to enhance the quality of features at each scale; then, the compressed excitation module is used to highlight useful features and obtain the relationship between features of different channels; finally, it uses the cross-stage local block to replace the continuous convolution, reducing the number of parameters and feature loss. The attention mechanism network EFNet (Liu et al., 2022) uses two eagle-eye foveal modules. The front one learns proposal object knowledge based on channel attention and spatial attention, while the back one uses two anchor-free sub-networks to predict refined objects. By combining attention information and using attention maps to generate adaptive anchor boxes, the model can better capture relevant information and optimize the detection process, thereby improving performance.

### 2.2 SMALL OBJECT DETECTION

Small object detection methods mainly focus on weak feature representation of small objects and lack of training samples for small objects. The two-stage method Faster R-CNN (Ren et al., 2017) introduces a Region Proposal Network (RPN), which first generates multiple proposal bounding boxes to provide a rough prediction of object location, and then further performs object classification and bounding box regression on these region proposals, with high accuracy and robustness. SSD (Single Shot Multibox Detector, Liu et al. (2016)) introduces a pyramid feature hierarchy and uses feature maps of different resolutions to predict objects of different scales. SSD ignores the complementary information between features at different levels, resulting in weak semantic information of low-level features. FPN (Lin et al., 2017a) introduces a top-down path to solve this problem, which transfers rich semantic information from deep features to shallow features, so that all levels of features contain rich semantic features and can generate high-quality feature representations at each scale. Inspired by FPN, DetectoRS (Yang et al., 2022) proposes a recursive feature pyramid, which merges the additional feedback connections of the feature pyramid network into the bottom-up backbone layer. For the problem of the imbalance between positive and negative object samples, RetinaNet (Zhang et al., 2020a) designs Focal Loss to solve the problem of class imbalance by reducing the loss weight assigned to well-classified samples. ATSS(Wang et al., 2021a) proposes an adaptive training sample selection strategy to automatically select positive and negative samples according to the statistical characteristics of the object, significantly improving the performance of anchor-based detectors and anchor-free detectors. NWD (Carion et al., 2020) points out that the Intersection over Union (IoU)-based metric is very sensitive to the position deviation of small objects, which will greatly reduce the detection performance. It also proposes a new evaluation metric called the normalized Wasserstein distance, which significantly improves the performance of small object detection.

Transformer-based detectors have embedded attention mechanism and strong context capture capability, making them suitable for small object detection. Deformable DETR (Li et al., 2020) designs a deformable attention module that only focuses on the interaction between the query element and a small group of sampled positions related to it. It also introduces a two-stage architecture, where

the last layer of the encoder stage generates predicted bounding boxes and selects the group with the highest score as region proposal. In the decoder stage, the region proposal is used to initialize the position embedding of the object query, allowing the attention module to focus on the object regions more quickly and accurately.

## 3 METHODS

Inspired by the two-stage Deformable DETR, a query optimization detection transformer for small object in remote sensing images is designed, called QO-DETR, and the overall architecture is shown in Fig.1. QO-DETR represents the query of the Transformer as a combination of 4-D dynamic anchor boxes and content features. Like Deformable DETR, QO-DETR uses multi-scale deformable attention in the self-attention module of the encoder and the cross-attention module of the decoder. Specifically, to address the problem of weak feature representation and uneven spatial distribution of small objects, QPG is designed to select queries based on the score of multi-class classification, provides initial position priors for the object query of the decoder, and provides position constraints for feature aggregation in the decoder attention module, so that the cross-attention module focuses on the object region, thereby enhancing the feature representation of the object and facilitating the location of the object region. To address the problem of high background noise in remote sensing images, GD is designed to introduce noise to candidate queries during training, enhance the network's ability to reconstruct object features from noise, reduce the interference of noise on bipartite graph matching, and improve the model's robustness to noise. In order to accurately locate small objects, QCR is designed for the decoder. Each decoder layer refines the query anchor box parameters layer by layer under the guidance of the adjacent layers to achieve spatial alignment between the anchor box and the object.

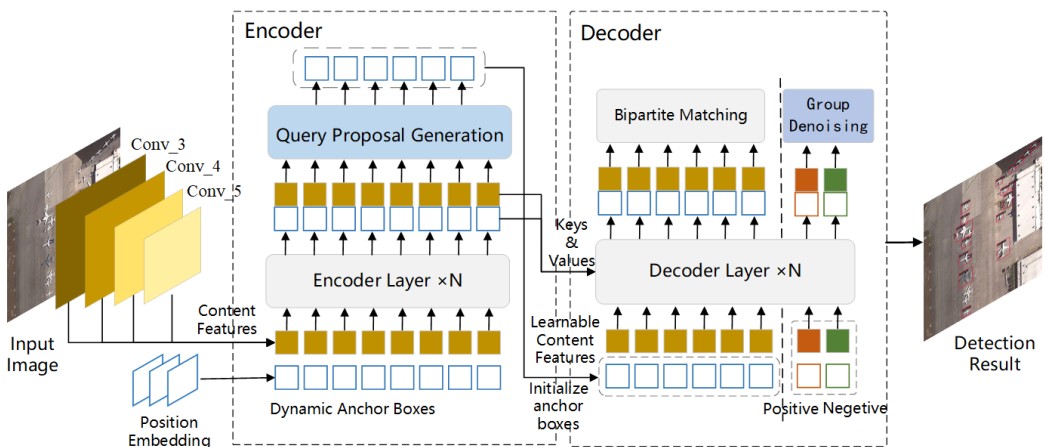

Figure 1: query optimization detection Transformer for small objects in remote sensing images

First, the ResNet backbone network extracts multi-scale feature maps for a given image and passes them into the Transformer encoder through position embedding. The encoder uses multi-scale deformable attention to enhance feature representation. Then, QPG calculates the multi-class classification scores of the anchor box queries output by the encoder and selects the anchor box queries with high scores as query proposals. The Transformer decoder uses these query proposals to initialize the position embedding of the object query, but does not initialize the content features of it. During training, GD treats the ground-truth bounding box of noise as a noisy query, and feeds it into the Transformer decoder together with the learnable anchor box query. For noisy queries, the reconstruction loss is used to guide the network to reconstruct their corresponding ground-truth bounding boxes; for learnable anchor box queries, the same training loss and bipartite matching as DETR are used. During inference, GD is not used. The Transformer decoder uses multi-scale deformable cross-attention to combine the features of the object query and the encoder output, and refines the query layer by layer in a cascade manner. Finally, the detection head predicts high-precision localization and classification results on the refined anchor box and content features.

## 3.1 QUERY PROPOSAL GENERATION

In DETR, the object query in the decoder is a static embedding without acquiring any encoder features related to the current image. The object query directly learns the position embedding from the training data without learning the content features, which makes each object query have no clear physical meaning and cannot explain where it will focus on the image. Therefore, DETR converges slowly during the training stage and has poor detection performance for small objects. To solve this problem, the two-stage Deformable DETR introduces a region proposal mechanism. Deformable DETR represents the Transformer query as a combination of 2-D reference points and content features. In the first stage, the last layer of the encoder uses an auxiliary detection head to generate coarse prediction boxes, and selects the prediction boxes with high scores as region proposals based on the binary classification (foreground and background) scores of the prediction boxes; in the second stage, these region proposals are fed into the decoder as the initial values of the object query. The decoder of the two-stage Deformable DETR uses region proposals to initialize the position embedding and content features of the object query at the same time. However, since these region proposals are obtained based on the binary classification scores, they can only represent the foreground but not the object instance, which may result in the situation where the content features contain multiple objects or only part of a object. Initializing the decoder's object query with these content features will affect the subsequent detection effect. To address this problem, a new QPG is proposed.

QO-DETR represents the query of Transformer as a combination of 4-D dynamic anchor box $(x, y, w, h)$ and content features, where $(x, y)$ is the center coordinate of the anchor box, and $w$, $h$ corresponds to the width and height of the anchor box, respectively. This representation method enables the query to focus on objects of different sizes, and QPG is shown in Fig.2. For the anchor box query output by the encoder, QPG calculates the multi-class classification score of each query and selects the queries with the top-K scores as query proposals. Then, the position embedding of the decoder object query is initialized by using the anchor box coordinates of the query proposal. In QO-DETR, it only initializes the position embedding of the object query, while the content features are not initialized. The content features of the object query are the same as DETR, which are learnable static content queries. This method helps the model use better position information to collect more comprehensive content features from the encoder, allowing the cross-attention module to focus on the local region corresponding to the object, thereby enhancing the feature representation of the object and helping the network to locate the object region more quickly and accurately.

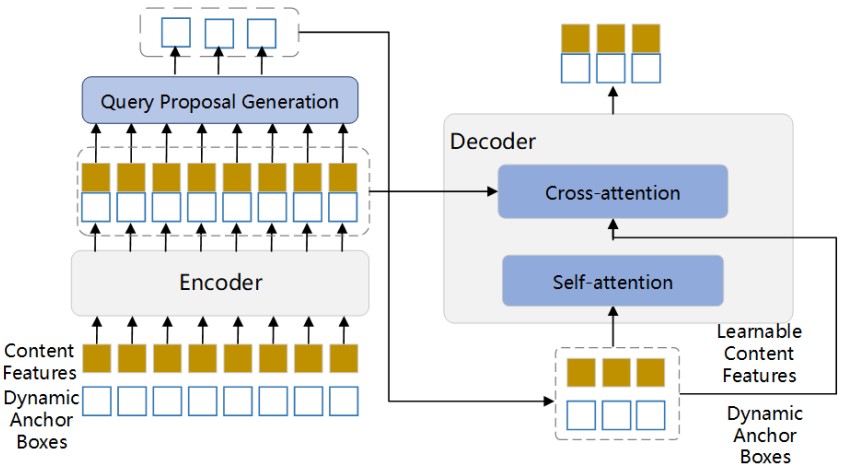

Figure 2: Query Proposal Generation

## 3.2 GROUP DENOISING

The training process of QO-DETR can be considered as two stages, learning anchor boxes and learning relative offsets. The query of the decoder is responsible for learning anchor boxes. The

inaccuracy of anchor box updates will make it difficult to learn relative offsets. Remote sensing images have complex backgrounds, and regional features of small objects are easily submerged by background and other instances, resulting in noise in the learned feature representations, affecting anchor box positioning. To address this problem, a denoising training method in QO-DETR is introduced to enhance the model's ability to reconstruct ground-truth features from noisy ones.

A noisy query represents an anchor box, which has a corresponding ground-truth bounding box nearby. The optimization goal of denoising training is to predict the ground-truth bounding box from the noisy query anchor box, which essentially avoids the ambiguity caused by Hungarian matching. The proposed Group Denoising module is shown in Fig.3. For each ground-truth bounding box, GD generates a positive query and a negative query, and uses two hyperparameters $\lambda_1$, $\lambda_2$ to control the scale of the noise, where $\lambda_1 < \lambda_2$. The positive query is inside the square, and the noise scale is less than $\lambda_1$, which is expected to be used to reconstruct its corresponding ground-truth bounding box. The negative query corresponds to the part between the inner and outer squares, and the noise scale is greater than $\lambda_1$, less than $\lambda_2$, which is expected to be used to predict the 'no object' category. QO-DETR uses multiple denoising query groups, each containing the same number of positive queries and negative queries. The reconstruction loss includes $l_1$ and GIOU loss for bounding box regression, and Focal Loss (Zhang et al., 2020a) for classification.

In addition, use an attention mask to prevent the leakage of noise information between matched parts and denoising groups, as well as between different denoising groups. Use $\boldsymbol{A} = [a_{ij}]_{W \times W}$ represent the attention mask, $a_{ij} = 1$ means that the $i$-th query cannot see the $j$-th query, and $a_{ij} = 0$ is the opposite. Then, the attention mask is expressed as:

$$
a_{ij} = \begin{cases} 1, & \text{if } j < P \times M \text{ and } \left[\frac{i}{M}\right] \neq \left[\frac{j}{M}\right] \\ 1, & \text{if } j < P \times M \text{ and i} \geq P \times M; \\ 0, & \text{otherwise.} \end{cases}
\tag{1}
$$

where $W = P \times M + N$, $P$ and $M$ are the number of denoising query groups and the number of ground-truth bounding boxes, respectively, and $N$ is the number of queries in the matching part. The first $P \times M$ rows and columns represent the denoising part, and the rest represent the matching part.

The proposed GD can suppress confusion and select high-quality anchor box queries to predict bounding boxes. When multiple anchor boxes are close to an object, it is difficult for the model to decide which anchor box to select, which may lead to two problems: the model makes repeated predictions and the model may select anchor boxes that are far away from the ground-truth bounding box. Denoising queries allow the model to distinguish subtle differences between anchor boxes and avoid repeated predictions. Since the essence of denoising training is to train the model to reconstruct bounding boxes from noisy anchor boxes that are close to the ground-truth bounding box, the model will search for predictions more locally, which makes each query focus on nearby regions, thereby preventing potential prediction conflicts between queries. It should be noted that this module is only used in training and removed during inference.

### 3.3 QUERY CASCADE REFINEMENT

To refine the anchor box query and achieve more accurate positioning of small objects, QCR is designed in the decoder to refine the anchor box query layer by layer in a cascade manner. For each decoder layer, the predicted bounding box of this layer is refined based on the refined box information of the first two layers and the prediction results of the previous layer, that is, the parameters of the $i$-th layer are affected by the losses of the $i$-th layer and the $i + 1$-th layer. Each predicted offset $\Delta b_i$ will be used to update the box twice, once for $b_i'$ and the other for $b_{i+1}^{pred}$. The final accuracy of the predicted box $b_{i+1}^{pred}$ is determined by two factors: the quality of the initial box $b_{i-1}$ and the predicted offset $\Delta b_i$ of the box. Given the input box $b_{i-1}$ of the $i$-th layer, the final predicted box $b_{i+1}^{pred}$ can be obtained in the following way:

$$
\begin{aligned}
\Delta b_i &= \text{Layer}_i(b_{i-1}), \quad b_i' = \text{Update}(b_{i-1}, \Delta b_i), \\
b_i &= \text{Detach}(b_i'), \quad b_{i+1}^{(\text{pred})} = \text{Update}(b_{i-1}', \Delta b_i)
\end{aligned}
\tag{2}
$$

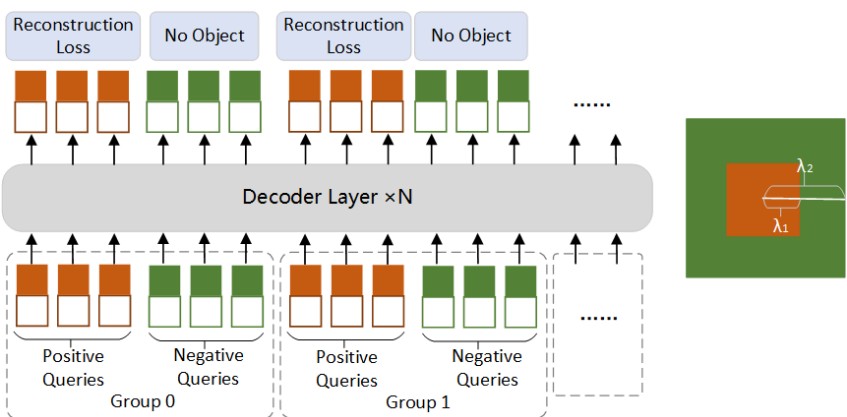

Figure 3: Group Denoising

where $\text{Update}\,(\cdot, \cdot)$ represents the refinement of the box $b_{i-1}$ by predicting the box offset $\Delta b_i$, and the specific process is:

$$b'_i = \left\{ \sigma \left( \Delta b_i^x + \sigma^{-1} \left( b_{i-1}^x \right) \right), \sigma \left( \Delta b_i^y + \sigma^{-1} \left( b_{i-1}^y \right) \right), \sigma \left( \Delta b_i^w + \sigma^{-1} \left( b_{i-1}^w \right) \right), \sigma \left( \Delta b_i^h + \sigma^{-1} \left( b_{i-1}^h \right) \right) \right\} \tag{3}$$

$\sigma$ and $\sigma^{-1}$ represent sigmoid and its inverse function respectively.

## 4 EXPERIMENTS

### 4.1 DATASETS AND EVALUATION METRICS

Use the DIOR (Li et al., 2020) and AI-TOD (Wang et al., 2021b) datasets to evaluate the proposed QO-DETR method. The definition of small objects follows the MS COCO dataset (Lin et al., 2014), and objects with an area smaller than $32 \times 32$ pixels are classified as small objects.

The DIOR dataset includes 23,463 images with a total of 192,472 instances, covering 20 object categories. Each instance is annotated by a horizontal bounding box. The image size is $800 \times 800$ pixels, and the spatial resolution is 0.5m~30m. The number of images in the training set, validation set, and test set are 5862, 5863, and 11738, respectively. More than 50% of the instances in the DIOR dataset are small objects.

The AI-TOD dataset includes 28,036 images with a total of 700,621 instances, covering 8 object categories. Each instance is annotated by a horizontal bounding box. The image size is $800 \times 800$ pixels. The number of images in the training set, test set, and validation set are 11214, 14018, and 2804, respectively. AI-TOD is specially designed for small object detection in aerial images, with an average absolute object size of only 12.8 pixels, and about 86% of the instances in the dataset are smaller than $16 \times 16$ pixels.

### 4.2 EXPERIMENTAL PARAMETER SETTINGS

ResNet-50 and ResNet-101 pre-trained on ImageNet are used as the backbone of the network, and multi-scale feature maps are extracted from conv3 ~conv5 of ResNet. A 6-layer Transformer encoder and a 6-layer Transformer decoder are used, the hidden feature dimension is 256, and the number of decoder queries is 900. The noise scale is $\lambda_1 = 1.0$, $\lambda_2 = 2.0$, and 100 denoising query groups are used, including 100 positive queries and 100 negative queries.

The initial learning rate (lr) is set to $1 \times 10^{-4}$, and lr is dropped at the 11th and 30th epochs for the training process of 12 and 36 epochs, respectively, using AdamW with a weight decay of $1 \times 10^{-4}$. Model training and inference are performed on a single GeForce RTX 3090 GPU, and the batchsize

of the training stage is 16. The parameters of the Focal Loss for classification are set to $\alpha = 0.25$, $\gamma = 2$. The weights of classification loss, L1 loss, and GIOU loss are set to 1.0, 5.0, and 2.0, respectively.

## 4.3 EXPERIMENTAL RESULTS

### 4.3.1 EXPERIMENTS ON THE DIOR DATASET

Table 1 shows the performance of the algorithm for detecting objects of different sizes in the DIOR dataset, and Table 2 shows the performance of the algorithm for detecting objects of different categories in the DIOR dataset. In the table, AP is averaged over all 10 IoU thresholds for all categories, ranging from 0.50:0.95 with a step size of 0.05. $AP_{50}$ and $AP_{75}$ are calculated over a single IoU threshold of 0.5 and 0.75 for all categories. The definition of small, medium, and large detection objects follows the evaluation metrics introduced in MS COCO, where S, M, and L represent small, medium, and large objects, respectively.

As shown in Tables 1 and 2, our QO-DETR achieves the best results on the DIOR dataset. When using the ResNet-50 backbone and training epoch 36, the AP reaches 51.3% and the $AP_{50}$ reaches 72.3%. Compared with the two-stage Deformable DETR, the AP is improved by 4.9 (10.5%) and the $AP_{50}$ is improved by 5.7 (8.6%). As shown in Table 1, compared with other methods, QO-DETR has significantly improved the performance in detecting small objects. Combined with Table 2, the vehicle and ship categories contain a large number of small objects. QO-DETR achieves the best performance in these two categories, further proving the excellent performance of the algorithm in the small object detection task. As shown in Table 1, QO-DETR is equally effective for large-sized objects and medium-sized objects. Combined with Table 2, QO-DETR achieves the best performance on 14 categories, which further shows that QO-DETR is very robust for the detection of multi-scale objects.

Table 1: Comparison of evaluation indicators on the DIOR dataset

*('TS TS-Deformable DETR' stands for two-stage Deformable DETR using iterative bounding box regression)*

| Methods | Backbones | AP | $AP_{50}$ | $AP_{75}$ | $AP_S$ | $AP_M$ | $AP_L$ |
|---|---|---|---|---|---|---|---|
| Faster R-CNN(Ren et al., 2017) | ResNet-50 | 43.5 | 63.1 | 45.8 | 7.1 | 26.8 | 54.4 |
| SDD(Liu et al., 2016) | VGG16 | 34.1 | 58.6 | - | 7.4 | 29.2 | 49.4 |
| YOLOv5+MFDF(Lv et al., 2022) | Darknet53 | 28.7 | 53.5 | - | 6.7 | 26.1 | 40.7 |
| EFNet(Liu et al., 2022) | ResNet-50 | 35.9 | 60.4 | 36.5 | 3.6 | 23.9 | 44.2 |
| TS-Deformable DETR(Zhang et al., 2020b) | ResNet-50 | 46.4 | 66.6 | 49.8 | 9.8 | 34.5 | 67.6 |
| QO-DETR | ResNet-50 | 51.3 | 72.3 | 55.6 | 13.4 | 39.9 | 72.6 |

### 4.3.2 EXPERIMENTS ON THE AI-TOD DATASET

Table 3 shows the performance of the algorithm for detecting objects of different sizes in the AI-TOD dataset, and Table 4 shows the performance of the algorithm for detecting objects of different categories in the AI-TOD dataset. The training epochs of QO-DETR are all 36. In the AI-TOD dataset, objects with an area between $2 \times 2$ and $8 \times 8$ pixels are classified as very tiny objects, objects with an area between $8 \times 8$ and $16 \times 16$ pixels are classified as tiny objects, and objects with an area between $16 \times 16$ and $32 \times 32$ pixels are classified as small objects. Objects with an area between $32 \times 32$ and $64 \times 64$ pixels are classified as medium objects, and there are no large objects. The subscripts VT, T, S, and M represent very tiny objects, tiny objects, small objects, and medium objects, respectively.

As shown in Tables 1 and 2, our QO-DETR achieved the best results on the DIOR dataset. When using the ResNet-50 backbone and training epoch 36, the AP reached 23.6% and the $AP_{50}$ reached 57.8%. Compared with DetectoRS using NWD, the AP increased by 13.5%, the $AP_{50}$ increased by 11.5, and the $mAP_{VT}$, $mAP_T$, and $mAP_S$ increased by 39%, 19.3%, and 1.7%, respectively. The performance in detecting very tiny, tiny, and small objects has been significantly improved. As

Table 2: Comparison of $mAP_{50}$ of various objects on the DIOR dataset

| Methods | Faster R-CNN (Ren et al., 2017) | SDD (Liu et al., 2016) | RetinaNet (Lin et al., 2017b) | PANet (Liu et al., 2018) | TS-Deformable DETR (Zhang et al., 2020b) | QO-DETR |
|---|---|---|---|---|---|---|
| Backbone Networks | ResNet-50 | VGG16 | ResNet-50 | ResNet-50 | ResNet-50 | ResNet-50 |
| Airplanes | 54.1 | 59.5 | 53.7 | 61.9 | 76.6 | 80.3 |
| Airports | 71.4 | 72.7 | 77.3 | 70.4 | 88.9 | 83.9 |
| Baseball fields | 63.3 | 72.4 | 69.0 | 71.0 | 71.7 | 73.9 |
| Basketball courts | 81.0 | 75.7 | 81.3 | 80.4 | 86.0 | 87.5 |
| Bridges | 42.6 | 29.7 | 44.1 | 38.9 | 42.8 | 45.6 |
| Chimneys | 72.5 | 65.8 | 72.3 | 72.5 | 76.2 | 82.7 |
| Dam | 57.5 | 56.6 | 62.5 | 56.5 | 58.0 | 70.1 |
| Highway service areas | 68.7 | 63.5 | 76.2 | 68.4 | 59.0 | 64.3 |
| Highway toll booths | 62.1 | 53.1 | 66.0 | 60.0 | 54.9 | 57.7 |
| Ports | 73.1 | 65.3 | 77.7 | 69.0 | 76.4 | 77.3 |
| Golf courses | 76.5 | 68.6 | 74.2 | 74.5 | 70.6 | 77.1 |
| Athletics fields | 42.8 | 49.4 | 50.7 | 41.6 | 50.5 | 61.8 |
| Overpasses | 56.0 | 48.1 | 59.5 | 55.8 | 58.4 | 61.2 |
| Shipboats | 71.8 | 59.2 | 71.2 | 71.7 | 75.4 | 90.3 |
| Stadiums | 57.0 | 61.0 | 69.3 | 72.9 | 56.2 | 71.2 |
| Storage tanks | 53.5 | 46.6 | 44.8 | 62.3 | 67.5 | 77.7 |
| Tennis courts | 81.2 | 76.3 | 81.3 | 81.2 | 84.0 | 85.2 |
| Train stations | 53.0 | 55.1 | 54.2 | 54.6 | 57.9 | 59.0 |
| Vehicles | 43.1 | 27.4 | 45.1 | 48.2 | 51.9 | 58.4 |
| Windmills | 80.9 | 65.7 | 83.4 | 86.7 | 76.6 | 77.7 |
| $mAP_{50}(\%)$ | 63.1 | 58.6 | 65.7 | 63.8 | 66.6 | 72.3 |

shown in Table 2, QO-DETR achieved the best performance on all classes, indicating that QO-DETR has excellent performance in small object detection tasks.

Table 3: Comparison of evaluation indicators on the AI-TOD dataset

| Methods | Backbones | AP | $AP_{50}$ | $AP_{75}$ | $AP_{VT}$ | $AP_T$ | $AP_S$ | $AP_M$ |
|---|---|---|---|---|---|---|---|---|
| RetinaNet(Lin et al., 2017b) | ResNet-50 | 4.7 | 13.6 | 2.1 | 2.0 | 5.4 | 6.3 | 7.6 |
| Faster R-CNN(Ren et al., 2017) | ResNet-50 | 11.4 | 27.0 | 8.0 | 0.0 | 8.3 | 23.1 | 24.5 |
| Cascade R-CNN(Cai & Vasconcelos, 2018) | ResNet-50 | 13.8 | 30.8 | 10.5 | 0.0 | 10.6 | 25.5 | 26.6 |
| SSD(Liu et al., 2016) | VGG-16 | 7.0 | 21.7 | 2.8 | 1.0 | 4.7 | 11.5 | 13.5 |
| ATSS(Zhang et al., 2020a) | ResNet-50 | 12.8 | 30.6 | 8.5 | 1.9 | 11.6 | 19.5 | 29.2 |
| M-CenterNet(Wang et al., 2021b) | DLA-34 | 14.5 | 40.7 | 6.4 | 6.1 | 15.0 | 19.4 | 20.4 |
| DetectoRS+NWD(Wang et al., 2021a) | ResNet-50 | 20.8 | 49.3 | 14.3 | 6.4 | 19.7 | 29.6 | 38.3 |
| TS -Deformable DETR(Zhang et al., 2020b) | ResNet-50 | 16.0 | 44.3 | 8.1 | 4.7 | 16.5 | 21.5 | 25.2 |
| QO-DETR | ResNet-50 | 23.6 | 57.6 | 15.8 | 8.9 | 23.5 | 30.1 | 35.2 |

Table 4: Comparison of $mAP_{50}$ of various targets on AI-TOD dataset

| Methods | Aircraft | Bridges | Tanks | Ships | Pools | Cars | People | Windmills |
|---|---|---|---|---|---|---|---|---|
| RetinaNet(Lin et al., 2017b) | 0.0 | 6.6 | 1.8 | 20.9 | 0.1 | 5.7 | 1.8 | 0.5 |
| Faster R-CNN(Ren et al., 2017) | 22.7 | 3.9 | 20.2 | 19.0 | 8.9 | 11.9 | 4.5 | 0.3 |
| Cascade R-CNN(Cai & Vasconcelos, 2018) | 25.6 | 7.5 | 23.3 | 23.6 | 10.8 | 14.1 | 5.3 | 0.0 |
| SSD(Liu et al., 2016) | 14.5 | 3.1 | 10.9 | 13.1 | 1.9 | 7.8 | 3.1 | 1.5 |
| M-CenterNet(Wang et al., 2021b) | 18.6 | 10.6 | 27.6 | 22.3 | 7.5 | 18.6 | 9.2 | 2.0 |
| TS-Deformable DETR(Zhang et al., 2020b) | 22.5 | 13.7 | 26.0 | 24.7 | 11.4 | 16.4 | 8.9 | 4.4 |
| QO-DETR | 32.0 | 16.7 | 35.7 | 39.8 | 18.0 | 22.6 | 12.7 | 6.3 |

### 4.3.3 ABLATION EXPERIMENT

Ablation experiments were conducted on the DIOR dataset to explore the effects of QPG, GD, and the decoder QCR on the detector performance. The results are shown in Table 5. The two-stage

Deformable DETR with decoder iterative bounding box regression is used as the baseline network, denoted as DeformDETR-TS-BBR. First, the region proposal module of the baseline network is replaced with QPG, denoted as DeformDETR-QPG-BBR. Then, GD is added, denoted as QO-DETR-BBR. Finally, the decoder iterative bounding box regression is replaced with the decoder QCR, which is QO-DETR. The training epochs of all models are 12.

As shown in Table 5, the AP is improved by 50.7% by replacing the region proposal part in the two-stage Deformable DETR with QPG; the AP is further improved by 0.6% by adding GD in the training stage; and the AP is further improved by 0.9% by using QCR in the decoder. The improvement in the detection performance of objects at each scale proves the effectiveness of the proposed method.

Table 5: Ablation experiment results on the DIOR dataset

| Methods | QPG | GD | QCR | AP | $AP_{50}$ | $AP_{75}$ | $AP_S$ | $AP_M$ | $AP_L$ |
|---|---|---|---|---|---|---|---|---|---|
| DeformDETR-TS-BBR | | | | 33.5 | 50.0 | 35.8 | 6.4 | 24.0 | 50.9 |
| DeformDETR-QPG-BBR | ✓ | | | 50.5 | 71.2 | 54.8 | 13.0 | 39.3 | 71.3 |
| QO-DETR-BBR | ✓ | ✓ | | 50.8 | 71.8 | 55.0 | 13.2 | 39.6 | 71.5 |
| QO-DETR | ✓ | ✓ | ✓ | 51.3 | 72.1 | 55.6 | 13.2 | 39.6 | 72.1 |

## 5 CONCLUSION

To solve the problem of high background noise and weak feature representation of small objects in remote sensing images, an end-to-end remote sensing image small object query optimization detection Transformer is designed, called QO-DETR. A Query Proposal Generation (QPG) module is designed to select queries according to multi-class classification scores, provide initial position embedding for the decoder's object query, and provide position constraints for feature aggregation in the decoder's attention module, so that the cross-attention module focuses on the object region, which can not only obtain features with stronger object representation capabilities, but also help the network locate the object region. A Query Cascade Refinement (QCR) strategy is designed to enable each decoder layer to refine the query anchor box parameters layer by layer under the guidance of subsequent layers, realize spatial alignment between the anchor box and the object, and accurately locate small objects. A Group Denoising (GD) module is designed to introduce noise to candidate queries during training, enhance the network's ability to reconstruct object features from noise, reduce the interference of noise on bipartite matching, and improve the model's robustness to noise. Experiments are conducted on large remote sensing datasets DIOR and AI-TOD. The AP on DIOR reaches 51.3% and the $AP_S$ reaches 13.4%. The AP on AI-TOD reaches 23.6% and the $AP_S$ reaches 30.1%, which proves the excellent performance of QO-DETR in small object detection tasks.

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
