# OpenReview forum: "Query Optimization Detection Transformer for Small Objects in Remote Sensing Images"
_ICLR.cc/2025/Conference — ICLR 2025 Conference Withdrawn Submission_

### Official Review · Reviewer_RJn4 · 2024-10-26

**Soundness:** 1
**Presentation:** 2
**Contribution:** 1
**Rating:** 3
**Confidence:** 5

**Summary:**

In this paper, the authors investigate a detection transformer(DETR) for small object detection in remote sensing images.

**Strengths:**

Investigating a better DETR-like method for small object detection in remote sensing images is meaningful and challenging.

**Weaknesses:**

## 1. The novelty.

The novelty of the proposed method is questionable, as it closely resembles DINO[1].

I did not find innovative designs compared to DINO. Since no supplementary material was submitted, anonymized code is necessary to verify these claims. Additionally, the paper does not cite DINO [1] or compare with it.

DINO[1] was based on DAB-Deformable-DETR[2], which was based on two-stage version of Deformable-DETR[3] and Conditional DETR[4].

### The `QPG` module (`Query Proposal Generation`) is similar to the `Mixed Query Selection` module of DINO[1].

> (this paper) QO-DETR represents the query of Transformer as a combination of 4-D dynamic anchor box (x, y, w, h) and content features.

> (this paper) For the anchor box query output by the encoder, QPG calculates the multi-class classification score of each query and selects the queries with the top-K scores as query proposals.

> (DAB-DETR[2]) We denote A_q = (x_q, y_q, w_q, h_q) as the q-th anchor, …… Given an anchor A_q, its positional query P_q is generated by: P_q = MLP(PE(A_q)).

> (Conditional DETR[4]) Each query is formed by adding a content query c_q (the embedding output from the decoder self-attention), and a spatial query p_q (i.e., the object query o_q).

> (DINO[1]) The top-K encoder features in the last layer are selected to initialize the positional queries for the Transformer decoder, whereas the content queries are kept as learnable parameters.

The DINO presents the code multi-class classification score for [topk selection](https://github.com/IDEA-Research/detrex/blob/c56d32e3d0262cff9835ebe80a0642965ae0cb3e/projects/dino/modeling/dino_transformer.py#L429-L436).

> (this paper) In QO-DETR, it only initializes the position embedding of the object query, while the content features are not initialized. The content features of the object query are the same as DETR, which are learnable static content queries.

> (DINO[1]) we only initialize anchor boxes using the position information associated with the selected top-K features, but leave the content queries static as before.

**The figure 2 is also similar to the figure 5(c) of DINO[1] paper.**

### The Group Denoising module is similar to the `Contrastive DeNoising Training` module of DINO[1]

> (this paper) For each ground-truth bounding box, GD generates a positive query and a negative query, and uses two hyperparameters λ1, λ2 to control the scale of the noise, where λ1<λ2. The positive query is inside the square, and the noise scale is less than λ1, which is expected to be used to reconstruct its corresponding ground-truth bounding box. The negative query corresponds to the part between the inner and outer squares, and the noise scale is greater than λ1, less than λ2, which is expected to be used to predict the ’o object’ category.

> (DINO[1]) we have two hyper-parameters λ1 and λ2, where λ1 < λ2.
> We generate two types of CDN queries: positive queries and negative queries. Positive queries within the inner square have a noise scale smaller than λ1 and are expected to reconstruct their corresponding ground truth boxes. Negative queries between the inner and outer squares have a noise scale larger than λ1 and smaller than λ2. They are are expected to predict “no object”.

**The figure 3 is also similar to the figure 3 of DINO[1] paper.**

### The QCR module (Query Cascade Refinement) is similar to `Look Forward Twice` module of DINO[1]

The formula 2 of this paper is similar to the formula 2 of DINO[1] paper.


[1] Zhang H, Li F, Liu S, et al. Dino: Detr with improved denoising anchor boxes for end-to-end object detection. [arXiv:2203.03605](https://arxiv.org/abs/2203.03605).

[2] Liu S, Li F, Zhang H, et al. Dab-detr: Dynamic anchor boxes are better queries for detr. [arXiv:2201.12329](https://arxiv.org/abs/2201.12329).

[3] Zhu X, Su W, Lu L, et al. Deformable detr: Deformable transformers for end-to-end object detection. [arXiv:2010.04159](https://arxiv.org/abs/2010.04159).

[4] Meng D, Chen X, Fan Z, et al. Conditional detr for fast training convergence. [arXiv:2108.06152](https://arxiv.org/pdf/2108.06152).

## 2. The writing quality.

The quality of the writing needs improvement, with several long paragraphs and repeated content.

Such as the second paragraph of ‘Introduction’ section.

## 3. The title relevance.

The relevance of the proposed method to 'remote sensing images' and 'small object detection,' as stated in the title, is weak. The method appears to be intended for general object detection.

## 4. Denoising effect

I disagree with the authors' claim that adopting a denoising strategy improves the model's robustness to feature noise caused by complex backgrounds in remote sensing images.

The denoising is not used for inference. So if the aforementioned opinion is right, why the feature noise is not considered during inference?

**Questions:**

- What are the differences between QO-DETR and DINO? The authors should provide reproducible code to validate these differences.

- Are there two types of loss functions?

> For noisy queries, the reconstruction loss guides the network to reconstruct their corresponding ground-truth bounding boxes. For learnable anchor box queries, DETR’s same training loss and bipartite matching are used.

DETR did not use Focal Loss.

> The reconstruction loss includes L1 and GIoU loss for bounding box regression, and Focal Loss (Zhang et al., 2020a) for classification.

The authors should clarify the exact formulation and purpose of each loss function, especially how they differ from DETR's original losses.

**Details Of Ethics Concerns:**

The authors should provide reproducible code to validate the contributions.

---

### Official Review · Reviewer_PrRP · 2024-11-01

**Soundness:** 1
**Presentation:** 1
**Contribution:** 1
**Rating:** 3
**Confidence:** 5

**Summary:**

The author proposed a DETR model for small object detection in remote sensing images, incorporating three targeted designs: proposal selection, group denoising, and query refinement. The model was evaluated on DIOR and AI-TOD datasets, achieving competitive performance in small object detection.

**Strengths:**

The structure of this article is well-organized, including essential sections such as methodology and experiments. Additionally, the experimental setup is fairly reasonable.

**Weaknesses:**

1. The article lacks innovation. Proposal Generation, Group Denoising, and Query Cascade Refinement have all been widely studied in works such as DINO, DAB-DETR, and Group DETR, yet the authors did not introduce any effective improvements. Additionally, these previous works are not referenced in the article. Using the DINO as an example, Figure 1 in this paper is highly similar to Figure 2 in the DINO. The content in the Mixed Query Selection section of the DINO paper closely aligns with Section 3.2 of this paper, with the only difference being that QPG in this paper selects proposals based on multi-class scores, whereas DINO uses the objectiveness scores. The Contrastive DeNoising Training section in the DINO is also nearly identical to Section 3.3 in this paper, with Figure 3 in the DINO paper and Figure 3 in this paper being almost the same. The relevant content for Equation 1 can be found in the Attention Mask section of DN-DETR. The Look Forward Twice section in the DINO paper is also largely similar to Section 3.3 in this paper, with identical formulas.
2. The comparative methods in the experiments are outdated, making it difficult to strongly validate the approach's effectiveness. The authors also did not compare their model with detection models like QueryDet, which are specifically designed for small object detection tasks. Some other baselines include CFINet (Small Object Detection via Coarse-to-Fine Proposal Generation and Imitation Learning), CEASC (Adaptive Sparse Convolutional Networks with Global Context Enhancement for Faster Object Detection on Drone Images), and DQ-DETR.
3. The experiments are severely insufficient. The authors only provided a table for the ablation study to analyze the effectiveness of the modules without further quantitative and qualitative analysis. For example, you could visualize detection results to qualitatively demonstrate the model's ability to detect small objects or visualize the recall performance of Proposal Selection for small object detection. Additionally, you could show the impact of the DeNoising module on the detector's loss function and so on.

**Questions:**

1. How does the performance of the QO-DETR compare with that of the DINO on small object detection tasks?

**Details Of Ethics Concerns:**

No Details

---

### Official Review · Reviewer_PLx7 · 2024-11-04

**Soundness:** 2
**Presentation:** 2
**Contribution:** 2
**Rating:** 3
**Confidence:** 4

**Summary:**

In this paper, the authors propose QO-DETR for small object detection in remote sensing images. The main modules include query proposal generation, group denoising, and cascade refinement strategy. These modules show some advantages in improving the accuracy of DETR.

**Strengths:**

The research topic is significant and the authors explore and improve DETR-based detection, achieving an accuracy higher than several well-known methods.

**Weaknesses:**

1. The related work is not comprehensive, e.g., A highly related work DINO [1] is not analyzed and compared. [1] https://arxiv.org/abs/2203.03605
2. The overall writing quality needs improvement. There are some errors, e.g., in Line 405 "TS TS-Deformable DETR". The caption of Fig. 1 is in lowercase, and the captions of other figures do not provide enough information.
3. The improvement calculation is misleading, e.g., in Line 491, "the AP is improved by 50.7%". Since AP is already a percentage value, I believe the better way is 50.5% - 33.5% = 17.0%.

**Questions:**

What's the difference between the proposed denoising and the denoising module in DINO?

---

### Official Review · Reviewer_U1Za · 2024-11-04

**Soundness:** 1
**Presentation:** 1
**Contribution:** 1
**Rating:** 1
**Confidence:** 5

**Summary:**

This paper introduces a DETR detection architecture for remote sensing images. For me, it looks like a large portion of its proposed method is plagiarised from previous works [a][b]. The model is tested on some remote sensing benchmarks.

 [a] Li, Feng, et al. "Dn-detr: Accelerate detr training by introducing query denoising." Proceedings of the IEEE/CVF conference on computer vision and pattern recognition. 2022.

[b] Zhang, Hao, et al. "DINO: DETR with Improved DeNoising Anchor Boxes for End-to-End Object Detection." ICLR 2023.

**Strengths:**

I cannot find any.

**Weaknesses:**

A large part of this paper is plagiarised from papers [a] and [b] (arxiv version). Some evidences are listed in the following.
1. This paper's section 3.2 is plagiarised from both [a] and [b]. Specifically, L275-L286 is plagiarised from the section 3.3 of [b]. L287-L299 is plagiarised from the section 4.4 of [a].
2. This paper's section 3.3 is plagiarised from section 3.5 of [b].
3. This paper's Fig.1 is actually the Fig.1 of [b].
4. This paper's Fig.3 is actually the Fig.3 of [b].
5. There is no mention or citation to [a] and [b].

[a] Li, Feng, et al. "Dn-detr: Accelerate detr training by introducing query denoising." Proceedings of the IEEE/CVF conference on computer vision and pattern recognition. 2022. (https://openaccess.thecvf.com/content/CVPR2022/papers/Li_DN-DETR_Accelerate_DETR_Training_by_Introducing_Query_DeNoising_CVPR_2022_paper.pdf)

[b] Zhang, Hao, et al. "DINO: DETR with Improved DeNoising Anchor Boxes for End-to-End Object Detection." ICLR 2023. (https://arxiv.org/pdf/2203.03605)

**Questions:**

N/A

**Details Of Ethics Concerns:**

Shown in weakness.

---

### Official Review · Reviewer_mDyx · 2024-11-05

**Soundness:** 1
**Presentation:** 1
**Contribution:** 1
**Rating:** 1
**Confidence:** 4

**Summary:**

This paper concerns the problem of remote sensing object detection. The designed model is named as QO-DETR, which consists of several modules such as Query Proposal Generation (QPG), Group Denoising (GD), and Query Cascade Refinement (QCR). Experiments are conducted on the DIOR and AI-TOD datasets.

**Strengths:**

Small object detection is a significant topic in the community of remote sensing image processing.

**Weaknesses:**

**I agree with Reviewer U1Za’s official comments that many of the technical designs in this paper have been proposed in some existing works. The authors state such ideas without appropriate citations.**


As introduced by Reviewer U1Za, the authors of this paper utilize the technical designs in the literature [a] and [b] without appropriate citations. I agree with Reviewer U1Za that the main technical designs of the Group Denoising (GD) module and Query Cascade Refinement (QCR) module have been proposed in Sections 3.3 and 3.5 of the literature [b]. In my opinion, the main idea of Query Proposal Generation (QPG) has also been introduced in Section 3.4 of the literature [b]. In the literature [b], Mixed Query Selection is designed to initialize anchor box queries using the position information associated with the selected top-K features while keeping the learnable static content queries. The Query Proposal Generation (QPG) module in this paper follows such a setting without citations.

If the main work of this paper is utilizing some existing technologies to solve the remote sensing object detection problem, the authors should clarify this fact and provide appropriate citations. I think it is hard to judge the real contributions of this paper in its current organization.

[a] Li, Feng, et al. "Dn-detr: Accelerate detr training by introducing query denoising." Proceedings of the IEEE/CVF conference on computer vision and pattern recognition. 2022. (https://openaccess.thecvf.com/content/CVPR2022/papers/Li_DN-DETR_Accelerate_DETR_Training_by_Introducing_Query_DeNoising_CVPR_2022_paper.pdf)

[b] Zhang, Hao, et al. "DINO: DETR with Improved DeNoising Anchor Boxes for End-to-End Object Detection." ICLR 2023. (https://arxiv.org/pdf/2203.03605)

**Questions:**

My questions have been introduced in the above weaknesses.

**Details Of Ethics Concerns:**

The details have been introduced in the above weaknesses.

---

### Note · Authors · 2024-11-13

I have read and agree with the venue's withdrawal policy on behalf of myself and my co-authors.